# Earlier Morning Arrival to Pollen-Rewarding Flowers May Enable Feral Bumble Bees to Successfully Compete with Local Bee Species and Expand Their Distribution Range in a Mediterranean Habitat

**DOI:** 10.3390/insects13090816

**Published:** 2022-09-07

**Authors:** Noam Bar-Shai, Uzi Motro, Avishai Shmida, Guy Bloch

**Affiliations:** 1Department of Ecology, Evolution and Behavior, The A. Silberman Institute of Life Sciences, The Hebrew University of Jerusalem, Jerusalem 91904, Israel; 2Jerusalem Botanical Garden, The Hebrew University of Jerusalem, Jerusalem 91904, Israel; 3Department of Statistics, The Hebrew University of Jerusalem, Jerusalem 91905, Israel; 4The Federmann Center for the Study of Rationality, The Hebrew University of Jerusalem, Jerusalem 91904, Israel

**Keywords:** *Apis mellifera*, *Bombus terrestris*, bumble bees, diurnal rhythms, Mediterranean habitat, native bees, pollen reward, pollination network, range extension

## Abstract

**Simple Summary:**

Many crops and wild plants depend on bee pollination for reproduction. Recent decades have shown evidence for a decline in the populations of many species of bees. One reason for this decline is the introduction of alien bees into new areas. We assessed the potential influence of bumble bees that are produced on an industrial scale and have continuously expanded their range in the Mediterranean climate regions of Israel. We found that all bee species in study sites in the Judean Hills in Israel tend to visit pollen-providing flowers at earlier times compared to nectar-providing flowers. Bumble bees and honey bees start foraging at earlier times and colder temperatures compared to native bee species. This means that the two species of commercially managed social bees are potentially depleting much of the pollen, which is typically non-replenished, before most local species arrive to gather it. Bumble bee ability to forage at the low temperatures of the early morning, and their capacity to collect pollen at early hours in the dry Mediterranean climate, potentially pose a significant competitive pressure on native bee fauna, and may lead to changes in the reproduction of local flower species.

**Abstract:**

During recent decades, bumble bees (*Bombus terrestris*) have continuously expanded their range in the Mediterranean climate regions of Israel. To assess their potential effects on local bee communities, we monitored their diurnal and seasonal activity patterns, as well as those of native bee species in the Judean Hills. We found that all bee species tend to visit pollen-providing flowers at earlier times compared to nectar-providing flowers. Bumble bees and honey bees start foraging at earlier times and colder temperatures compared to other species of bees. This means that the two species of commercially managed social bees are potentially depleting much of the pollen, which is typically non-replenished, before most local species arrive to gather it. Taking into consideration the long activity season of bumble bees in the Judean hills, their ability to forage at the low temperatures of the early morning, and their capacity to collect pollen at early hours in the dry Mediterranean climate, feral and range-expanding bumble bees potentially pose a significant competitive pressure on native bee fauna. Their effects on local bees can further modify pollination networks, and lead to changes in the local flora.

## 1. Introduction

Many wild and cultivated plants depend on animal pollinators for their reproduction [1,2,3]. Wild bees provide vital pollination services to natural and agricultural ecosystems, and therefore have critical influence on both biodiversity and economy [4]. Studies in diverse locations have repeatedly suggested that wild populations of many pollinators have been severely decreasing over past decades, accompanied by a decrease in, and even extinction, of their forage plants, a syndrome coined “the global pollination crisis” [5,6,7,8]. Many contributing factors for the pollination crisis have been suggested [9], one of which is the introduction of managed honey bees and bumble bees into areas outside their natural range [10,11,12,13].

*Bombus terrestris* is the bumble bee most commonly used for commercial pollination [14]. It is an annual species in which colonies are naturally founded by a single overwintering mated gyne. The social phase of their life cycle starts with the emergence of workers from the first batch of queen-produced brood, and colonies typically reach a maximal size of 200–300 workers and several dozen males and gynes [15]. Its native distribution range extends from most of Europe’s area (South of lat. ~60.00° N), and the Mediterranean region to western Asia [16,17]. Since the 1980′s *B. terrestris* colonies have been artificially produced in industrial scale and used world-wide for commercial crop pollination, specifically in greenhouses [14]. The introduction of *B. terrestris* colonies into new countries and areas outside its natural range has led to a notable increase in its global distribution with reports of feral or invasive populations in places such as Turkey, Japan, South America, New Zealand, and Tasmania [15,18].

In Israel, the original distribution of *B. terrestris* (subsp. *dalmatinus*) was limited to the hills and mountains of the Upper Galilee and Mt. Hermon [19] in the northern parts of the country (lat. ~33.00° N), which lies at the southern margins of its global distribution. Over the last decades, *B. terrestris* has been observed to constantly expand its range towards southern Israel ([18,20], the authors’ unpublished observations). A long-term study showed a significant increase in the frequency of visits made by *B. terrestris* to *Arbutus andrachne* flowers since 1984, when it was first recorded at Mt. Carmel (lat. 32.76° N), up to 1994 [20]. *B. terrestris* was first observed at Mt. Gilo, south of Jerusalem (lat. 31.72° N) in 1995 (the authors’ unpublished observations), and on 2013 at Hebron area (lat. 31.48° N, [21]). This range expansion is attributed to both the increase in public and domestic garden areas, which provide ample floral resources throughout the year, and the increase in the commercial use of bumble bee colonies for crop pollination which may have led to unintended establishment of feral colonies [20].

Several characteristics of *B. terrestris’* biology are thought to contribute to its efficient spread and success as an invasive species. These include the capacity to exploit diverse habitats, broad foraging preference (polylectic foraging that includes garden plants), flexible nesting preferences, and a relatively high reproductive capacity, which results in a demand for much food (reviewed in [15,18]). Their invasive potential is further enhanced by good dispersal ability [15,18,22]. Bumble bees are highly generalist and are therefore likely to pollinate exotic invasive plants, which might outcompete local plants [23,24]. Many species are known to rob nectar by punching flowers without transferring pollen [25] which may reduce the amounts of nectar available to “legitimate” visitors [26,27].

The concern that range-expanding or introduced bumble bees stress the local bee faunas is enhanced by reports from multiple locations, including Japan [28], Tasmania [29], Chile, Argentina [30] and Israel [20]. Their range expansion might also influence plant communities because plants that are efficiently pollinated by *B. terrestris* might benefit from an increase in pollination, whereas plants that depend on pollinators which are outcompeted by *Bombus* may face reduced reproduction [31].

Here, we investigated the seasonal and diurnal foraging patterns of *B. terrestris* bumble bees (assumed to be feral) and native bees in the Judean Hills, Israel. This area is characterized by a warm semi-arid Mediterranean climate, which differs from the typical temperate climate throughout most of *B terrestris*’ native distribution, as well as most invasion localities reported for this species. *B. terrestris*, as most bumble bee species, is adapted to cold climate temperatures, and hence forage at lower ambient temperature and at earlier parts of the day [15,32] relative to most bees. Accordingly, we hypothesized that bumble bees arrive to flowers early in the day and reduce, or even deplete entirely, floral resources before the arrival of native Mediterranean species, which are presumably less capable of active foraging at low temperatures. We assume that reward depletion might be more pronounced for pollen resources than for nectar, which in many species can be continuously replaced during the flower blooming period [2,33], contrary to pollen, which is not typically produced to compensate for depletion [34,35,36]. In order to address possible differences between foraging for nectar and pollen, we compared visits to flowers from which bees collect mostly pollen with visits to flowers from which they collected mostly nectar rewards.

## 2. Materials and Methods

We performed observations during 2012–2014. The first year (2012) was devoted to characterizing the yearly and daily times during which *B. terrestris* bees are active, and the common flowers they visit in our study sites. We recorded the number of visits and identity of flower species visited by *B. terrestris* as well as by other bee species.

We selected two ~700 m long transects: one adjacent to the Kennedy Memorial and Se’adim Ruins (31°45′02.6″ N 35°08′04.9″ E, ELEV. 740–820 m; Figure 1), and the second at the upper Soreq Valley (31°46′08.0″ N 35°08′20.7″ E, ELEV. 510 m). Both sites are located 2–3 km W of Jerusalem (Figure 1). Climate in this area is Mediterranean with dry summers and an average annual rainfall of 550–600 mm [37]. Natural vegetation is low Mediterranean shrubland (i.e., Batha and Garigue), but there is some disturbance in both sites because of roads passing through them. There were cultivated gardens within a distance of less than 1 km from both sites, at the Kennedy Memorial and at the villages of Aminadav and Even-Sapir. Introduced plants that grow within the transects, alongside the local flora, are *Antirrhinum*
*majus* subsp. *tortuosum* and *Spartium junceum*, both are native in northern Israel, and *Centranthus ruber* which is not native in Israel. Among the plants mentioned above, bumble bee workers and males were observed visiting only *Antirrhinum*. A few bumble bee queens were observed visiting *Spartium* flowers, but these observations were not during our sampling activity. We detected managed honey bee hives in a distance of several hundred meters from the Soreq site. Feral honey bee colonies are known to exist in the Judean Hills, but were not seen by us in the vicinity of our research areas.

The sites were visited every few days from the beginning of March 2012 until the first appearance of bumble bees (only *B. terrestris* is found in this area) at the end of March. After recording the first bumble bees, we initiated a regular sampling schedule in which we visited the sites once every two weeks. We visited the sites until bumble bee activity ceased, towards the end of August. During 2012, we performed sampling walks along the transects at 6:00, 9:00, 12:00, and 18:00. At each sampling day, we recorded all blooming plant species along the transects. At the beginning of each sampling walk, we recorded the ambient temperature using Casella mercury-filled thermometer. Sampling was performed by walking along the transect and recording every *B. terrestris* individual that was observed visiting a flower, and the species of the plant. We defined a visit as an event in which a bee was observed physically touching the parts where floral rewards (nectar and/or pollen) are present. We aimed to record the caste and gender of the bee (queen, male or worker), but this was not always possible (specifically for unambiguously distinguishing males from workers). We also recorded any other bees visiting these flowers. The other bee visitors were assigned to one of three categories: (1) honey bees (which includes one species—*Apis mellifera*), (2) carpenter bees (mostly *Xylocopa violacea*, and less commonly *X. pubescens and X. iris*), (3) all other wild bees that are not assigned to one of the three groups mentioned above. Non-anthophilans visitors, such as dipterans (hover flies—Syrphidae), coleopterans, lepidopterans, and birds (sunbird—*Nectarinia osea*), were rarely seen and are not included in our analyses.

Based on the data obtained during 2012, which revealed that the highest bumble bee activity occurs in the morning (Figure 2), we decided to focus our observations at these hours. We also decided to limit data collection to the upper Soreq site, given that the other site was adjacent to the Kennedy Memorial gardens, which apparently attracted visitors from the adjacent natural areas, including our sampling site, hence reducing their numbers. Accordingly, for the field seasons of 2013 and 2014 we concentrated on the transect at the upper Soreq Valley. During these two years, we sampled along the transect every two weeks from April to late August. During 2013, samplings along the transect were performed every hour, starting at around sunrise, until when *B. terrestris* activity has ceased at around 11:00. During 2014, we concentrated our efforts at the earlier hours, during which we already knew that bumble bees are more commonly observed, and sampled from a little before sunrise until 9:00 AM. During both years, data were recorded using the same procedures as described for 2012. Altogether, we visited the site over 13 mornings comprising 65 transect scans in 2013, and 10 mornings, comprising 22 transect scans in 2014.

Flowering plant species were categorized according to rewards collected by visiting bees to be either nectar-rewarding or pollen-rewarding flowers (Table 1). This categorization was done by observing the behavior of bees while visiting the flowers. Nectar collection was characterized by the bee hovering or standing, with the proboscis extended and inserted into the known or assumed location of the nectar. In many cases, we were able to clearly see abdominal contractions (“pumping” movements), but this was not imperative for the classification. Pollen collection was characterized by the bee observed to move its legs while touching the anthers and grooming their pollen. The proboscis was usually folded during pollen collection. According to our observations, for each plant species, the same reward type was usually collected by all anthophilan visitors. In cases where we observed a main reward type in a certain plant species and an infrequent collection of the other reward type, (i.e., *Capparis zoharyi* and *Vitex agnus-castus*) we noted this in Table 1. Similar approaches of categorizing flowers to be either pollen- or nectar-providing flowers are commonly used in pollination ecology studies [1,34,35,38,39,40,41,42,43,44,45,46].

To assess the current state of *B. terrestris*’ distribution in the Judean hills, we further performed a few survey trips in which we recorded bumble bees’ presence during the time of their expected peak colony size (Appendix A).

## 3. Statistical Analysis

In accordance with our preliminary observations (2012) and earlier studies, as mentioned in the Introduction, we set to test two hypotheses: (1) Pollen foraging precedes nectar foraging; (2) bumble bees can perform efficient foraging earlier and in lower temperatures than the local species. The directionality of these two hypotheses allowed us to use one-tailed tests for our statistical analyses.

We analyzed each of the two years 2013 and 2014 separately, and the *p*-values were then combined to obtain an overall significance level, by applying the Mosteller and Bush method of adding weighted *z*’s [47]: Let *p*_1_ and *p*_2_ be the one-tailed *p*-values for 2013 and 2014. We denote by *z*_1_ and *z*_2_ the corresponding standard normal values, using the one-to-one mapping of the standard normal curve to the *p*-value of a one-tailed test. The combined weighted *z* will be
$$z_{total}=\frac{df_{1}\times z_{1}+df_{2}\times z_{2}}{\sqrt{df_{1}^{2}+df_{2}^{2}}}$$
where *df*_1_ and *df*_2_ are taken to be the number of scanning transects performed on each year. The combined weighted *p*-value is calculated using $z_{total}$.

Let *N* be the number of transect scans performed during a year. Our sampling units were the various transect scans, each characterized by the degree of the ambient temperature and by the time since sunrise at its start. Each of the *N* transect scans can be considered as a vector having eight components: the number of each of the four bee categories (bumble bees, honey bees, carpenter bees, and other wild bees) on nectar flowers, and the number of these four groups on pollen flowers:$$(X_{1,k},X_{2,k},X_{3,k},X_{4,k},Y_{1,k},Y_{2,k},Y_{3,k},Y_{4,k})k=1,\ldots ,N.$$

For standardization, we converted these numbers into proportions
$$(x_{1,k},x_{2,k},x_{3,k},x_{4,k},y_{1,k},y_{2,k},y_{3,k},y_{4,k}),\;$$
where
$$x_{1,k}=\frac{X_{1,k}}{\sum_{i=1}^{N}X_{1,i}},$$
etc.

These eight variables do not fit normal distribution, and observations on individual bees are not independent (i.e., bees were clustered within scans, and scans within different days). Thus, we had to apply a non-parametric approach for the statistical analyses as detailed in the following sections.

### 3.1. Dependence of Flower Visitation on the Time after Sunrise

For each of the eight combinations of bee categories (bumble bees, honey bees, carpenter bees, and other wild bees) and flower type, we calculated the mean time after sunrise (defined as the moment the upper edge of the solar disk—called the upper limb—becomes visible above the horizon) for each visitation record. We then used these data for three types of comparisons. (1) We compared the time of visitation records on all flowers of bumble bees to that of each of the three other bee categories. (2) For each bee category, we compared the time of visitation on nectar and pollen flowers. (3) For each flower type separately (nectar or pollen), we compared the mean time of visitation of bumble bees to that of each of the three other bee categories. These comparisons were done in the following way: First, we calculated the difference between the relevant components at each standardized vector, i.e.,
$$(x_{j,k}+y_{j,k})-(x_{1,k}+y_{1,k})\;j=2,3,4\;k=1,\ldots ,N$$
for the first type of comparisons;
$$x_{j,k}-y_{j,k}\;j=1,\ldots ,4\;k=1,\ldots ,N$$
for the second type of comparisons;
$$x_{j,k}-x_{1,k}\;j=2,3,4\;k=1,\ldots ,N$$
and
$$y_{j,k}-y_{1,k}\;j=2,3,4\;k=1,\ldots ,N$$
for the third type of comparisons. Next, we multiplied each difference by the corresponding time elapsed after sunrise of that vector, and summed up all these *N* products, to obtain the critical value for each of the 10 comparisons. The significance of each comparison was calculated by computer generated permutations. For each case, 10^4^ random permutations of the *N* time values were generated. Each permutation produced a corresponding sum of products that was compared to the critical value, and the proportion of permutations having a sum of products larger than, or equal to the critical value, served as an estimate of the *p*-value.

### 3.2. Dependence of Flower Visitation on Ambient Temperature

We performed similar analyses to those described above for the influence of time after sunrise, but using the recorded ambient temperature as the independent variable instead.

## 4. Results

### 4.1. Bombus terrestris Phenology in the Judean Hills

Combining our observations from both sites, in all three years, bumble bee activity was observed between the end of March until the beginning of September. At all three years, bumble bee workers were observed for the first time at the end of March or early April, and peak numbers were typically recorded during the first half of June (Figure 3). Bumble bees were recorded visiting flowers of many plant species, which varied with the progression of the season (Appendix A). We noted that on most plant species they were collecting either pollen or nectar, but rarely both types of reward (Table 1). All plant species visited by bumble bees were also visited by other bee species of various sizes and taxonomic groups, setting the stage for interspecies competition.

### 4.2. Flower Visitation during the Day

During 2012, we recorded bee activity every three hours from 6:00 am to noon plus an evening observation at 18:00, allowing us to assess the daily patterns of activity. The daily activity pattern was typically bimodal with many bumble bees observed at early morning, just after sunrise, then number of bumble bees observed declined towards noon, and was high again towards dusk (Figure 2). Flower visitation rates around noon were low for all four bee groups and were increased again at the evening. The observations during 2012 hinted for possible differences in the daily profile of activity for the four bee categories. The bumble bees and the honey bees tended to show the highest level of activity during the first morning observation (around 6:00) and again during the single evening observation (around 18:00), whereas *Xylocopa spp.* were recorded at similar numbers throughout the day, and the species grouped in the other native bees category showed peak activity around 9:00 (Figure 2). The earlier morning arrival of the bumble bees and managed honey bees could enable them to deplete flower resources before the arrival of the local wild species. In order to better assess the effect of time of day, we focused our 2013 and 2014 observations on the morning hours, and increased their resolution.

### 4.3. Dependence of Flower Visitation on the Time after Sunrise

The higher resolution of observations during 2013 and 2014 strengthen the evidence for early arrival of bumble bees (and honey bees) to flowers, with an overall similar pattern in both years. To make our observations ecologically relevant, we analyzed visitation rates relative to the time of sunrise. We found that bumble bees and honey bees arrive to flowers shortly after, or even before sunrise, and earlier than the two categories of wild bees. Our statistical analyses revealed a significantly earlier arrival time of bumble bees compared to carpenter bees and the general group of other wild bees, but not compared to honey bees (Table 2; Figure 4 and Figure 5).

Given that nectar, but not pollen, can be rapidly replenished [34,35], we further separately analyzed visits to flowers providing mostly nectar and visits to flowers providing mostly pollen. We found that the visitation rates increased earlier on pollen-rewarding flowers for all four bee categories (Table 3). Bumble bees visited both pollen- and nectar-rewarding flowers earlier than the wild bees, and the pollen-rewarding flowers earlier than the carpenter bees, but with no significant difference compared to the honey bee for both type of flowers (Table 4).

In a set of complementary analyses, we focused only on the first hour after sunrise. We compared the flower visitation records of bumble bees and that of each of the other three groups, using computer-generated 10^4^ random permutations for each comparison. In these analyses, we found that bumble bees are significantly more likely to be found on pollen-rewarding flowers compared to both the carpenter bees and the group of the other wild bees (*p* < 0.001 for each of the two groups; results combined for both years). The comparison to honey bees was statistically significant in 2014, but not in 2013 or in the pooled analyses of the two years together (Table 5). On nectar-providing flowers, the presence of bumble bees during the first hour after sunrise was not higher than for the other bee categories. Collectively, these analyses show that bees from all four categories tend to visit pollen-rewarding flowers earlier than they visit nectar-rewarding flowers. Bumble bees (and to lesser extent honey bees) arrive to pollen-rewarding flowers earlier in the morning and have the opportunity to deplete pollen rewards before the later arrival of the local wild bees (*Xylocopa* and other wild bee taxa).

### 4.4. Dependence of Flower Visitation on Ambient Temperature

To assess the influence of ambient temperatures more precisely, we reanalyzed our flower visitation records for each of the four bee categories according to the temperature recorded for each flower visitation event (i.e., beginning of transect). Overall, bumble bees were recorded on flowers at lower ambient temperatures compared to the carpenter bees and the category of the other local wild bees (Figure 6; Table 6). Bees from all four categories showed lower mean temperature for the records on pollen compared to nectar-rewarding flowers (Table 7). When we compared bumble bees to the other three categories, we found that they were recorded at lower ambient temperatures compared to wild bees. We found this trend for both the analyses of pollen- and nectar-rewarding flowers (Table 8). These analyses suggest that bumble bees can forage at lower temperatures than the local wild bees, allowing them to arrive earlier in the morning to pollen-rewarding flowers.

## 5. Discussion

One of the factors that may negatively impact natural bee populations is the spread of managed or commercially distributed species, predominantly honey bees and bumble bees, which are introduced into new areas in large numbers. The range expansion of *Bombus terrestris* has been recorded in Israel since the 1990’s [20], and is correlated with their commercial production, the increase reliance on this species for crop pollination, and the rise in garden areas. Similar range expansions or invasions have been reported in other countries [18,48,49]. We show that *B. terrestris* has successfully expanded its geographical distribution into the semi-arid Judean hills. Gynes were typically observed at the early spring (the authors’ personal observations), and workers were seen on flowers from early/mid-April until late August/early September, with peak population size recorded on late June. The strong seasonal pattern, and the fact that many of these bumble bees were recorded in locations that are not adjacent to commercially pollinated crops, suggest that these bees represent locally established populations. The introduction of bumble bees (due to either anthropogenic or spontaneous range expansion) into new areas can potentially reduce resources available to native flower-visiting species, and may have substantial influence on the local fauna and flora.

Our results suggest a previously overlooked mechanism that may amplify the competitive ability of bumble bees and facilitate their range expansion in Mediterranean climates. We found that bumble bees foraged at lower ambient temperatures and started foraging at earlier hours than the local native bees. This advantage is particularly notable relative to bees other than *Apis*, and *Xylocopa* which were recorded visiting flowers at significantly higher temperatures and later time of the day, compared to the bumble bees (Figure 3, Figure 4, Figure 5 and Figure 6). Our observations suggest that their early arrival is functionally significant because the bumble bees first visit pollen-rewarding flowers, and diminish, and perhaps even entirely deplete their pollen reward. This is specifically significant since pollen is not known to be replenished within the flower [34,35]. Thus, later arriving native bees may face shortage in pollen, which is the principal protein source for their brood [35,46]. Although it is well established that many bumble bee species are adapted to a cold, temperate climate, to the best of our knowledge, our study is the first to clearly show that this adaptation potentially provides them with competitive advantages in warmer and drier Mediterranean climate, where the pollen is usually dry and collectible at the early morning hours.

Several traits of bumble bees could account for their ability to visit flowers earlier during the day, even before sunrise. 1. Bumble bee physiology is apparently adapted to cold climates in which they overwinter and forage under low ambient temperatures [15,32,50]. 2. Bumble bees can actively raise their body temperature by shivering-like activation of their large throatic flight muscles [51,52]. 3. They socially thermoregulate and insulate their nests so that individual foragers need to spend less time and energy increasing their body temperature before their first morning flights. 4. Bumble bees show profound size polymorphism, with the larger bees typically foraging outside [53]. Their large size, relative to most local native bees, enables better heat retention [51] and better vision under low-light intensity [54]. A possible exception is the native “twilight bee” *Xylocopa olivieri* which specializes in foraging under very low light intensity during dusk and dawn hours [55], and hence start foraging even earlier than bumble bees. However, we observed only a few individuals of this species throughout the three-year study duration and excluded them from our dataset. Our finding that the highly social and thermoregulatory honey bees were also seen on flowers early in the morning (Figure 3, Figure 4 and Figure 5), suggests that the local native solitary species might be facing a growing stress from both bumble bees and managed honey bees that have been introduced into their natural habitats.

Considering their earlier arrival to pollen-rewarding flowers, and their higher abundance during early morning hours and low ambient temperatures, bumble bees can have competitive advantage over many native, specifically the non-Xylocopan, local bee species, which typically arrive to pollen-providing flowers later during the day, when less pollen remains at the flowers. We did not classify the many species of local bees in our study site, but it is reasonable to assume that some species suffer more than others from competition with bumble bees. Assuming that some wild bee populations are resource limited [56], the possible competition with early arriving bumble bees (as well as honey bees) may imply that some native bees need to work harder and invest more time and energy in order to collect the proteins (i.e., pollen) needed for provisioning their offspring. A shortage in pollen means less offspring and reduced fitness, even when nectar availability is sufficient [46,56].

Earlier arrival to pollen-rewarding, compared to nectar-rewarding flowers, was seen for bees from all four categories (Table 2). Similar results were found in several studies that were conducted in Tropical America [57,58,59,60], Pennsylvania [46] as well as Sinai [61]. This seems to be a beneficial strategy, given that pollen is typically a non-replenished resource, whereas nectar can be continuously replaced in many plant species. Nevertheless, it is not clear how general the preference is to visit pollen-rewarding flowers early in the morning. For example, a different pattern was reported in England for *B. terrestris*, which collected nectar at the earlier colder hours, and pollen at the later warmer hours of the day [62]. The authors attributed this finding to the humid temperate climate at the study area which made pollen collection difficult because of the high early-morning humidity in their study sites. In an earlier study, Shelly et al. [63] found that the desert-dwelling *Bombus pennsylvanicus sonorous* collected pollen at earlier hours, compared to nectar. This finding is consistent with our suggestion that early pollen foraging is possible in dry climates, where the pollen is in a collectible state even at the early hours. It is worth mentioning that Shelly et al. [63] and Peat and Goulson [62] did not compare the pollen collection hours of bumble bees with those of sympatric solitary bee species. Additionally, by contrast to our study, these studies were not conducted in places where bumble bees are relatively new alien species.

Bumble bee introduction into new areas can affect not only the native bees, but also the local flora, for several reasons. First, their assumed negative effects on native bees may change the structure of pollination networks, leaving some native flowers with less effective pollination due to the decline of their natural pollinators [6]. Second, alien bumble bees might visit and pollinate exotic plant species which are suitable for them, hence promoting the spread of invasive plants, which in turn may outcompete native plants, or further modify pollination networks [23,24]. Such a plant which was included in our study is *Antirrhinum majus* subsp. *tortuosum* which is an introduced plant in the Judean Hills, and was visited only by large bees, i.e., bumble bees and carpenter bees. The increasing number of bumble bees might facilitate its expansion. Third, the large numbers of bumble bees observed visiting flowers, specifically during the late spring and the summer, may give reproductive advantages to plants flowering during this time of year and being attractive to bumble bees at the expense of plants that bumble bees are less likely to pollinate. Fourth, bumble bees have the capacity to rob nectar from flowers of native plants (i.e., when the corolla tube is too long for their proboscis) making them less attractive or less rewarding to local native bees which possibly provide better pollination service by legitimately visiting these flowers [25,26,27].

While checking the distribution of plants that bumble bees visited in this study (Table 1 and Figure 7, [64,65]), it appears that all of them grow also north of Israel (at least at eastern Mediterranean Basin and/or southern Europe), in areas which lie within the presumed original native distribution of *B. terrestris dalmatinus* [16,17]. This may mean that bumble bees in the Judean hills have plenty of plants with which they co-evolved. This may differ from other areas into which bumble bees are introduced and which lie far away from their native range [18]. However, the significant of this speculative proposition needs to be further explored.

The exclusion, or even near-extinction, of native bumble bee species by invasive bumble bee species has been reported in several locations, including Japan [40,66], Chile, and Argentina [67,68,69]. However, this effect might be stronger when both interacting species are bumble bees, because of the high overlap in food niches, as well as the competition for nesting sites—which may be the main reason for such exclusions [70]. Reduction in solitary bees’ visitation in correlation with increased introduced bees’ activity (mainly honey bees), has been reported in many studies (e.g., [20,71,72,73,74,75]. While recording from the beginning of bees’ daily activity, at different hours and temperatures, and by comparing pollen to nectar flowers, our results suggest that the potential effect of competition comes not only from sharing the same resources, but also from the introduced bees’ capability to arrive earlier to the field and deplete the non-replenishable pollen. Future studies should aim for a finer taxonomic classification of the local bees. This will enable identification of species which are most vulnerable to the early visitation effects of bumble bees and honey bees.

## 6. Conclusions

*Bombus terrestris* is potentially a strong competitor of native bees in Mediterranean conditions. Our observations suggest that their competitive ability is potentially enhanced by their capacity to be active at lower temperatures and earlier hours of the day, relative to most native bees. These capabilities enable bumble bees (and honey bees, to a lesser extent) to arrive first to pollen-rewarding flowers and deplete the non-replenishable pollen resource. We suggest that in dry climates, pollen is easily collectible even at early hours. Many native bee species can potentially suffer from such strong competition throughout the long activity season of *B. terrestris* in this area. Additional studies with finer taxonomic resolution are needed to assess the effects of early foraging managed or introduced bees on specific local species.

The distribution records we obtained in this study at the presumed expansion front of *B. terrestris* may help in assessing the degree and rate of this expansion in future surveys.

## Figures and Tables

**Figure 1 insects-13-00816-f001:**
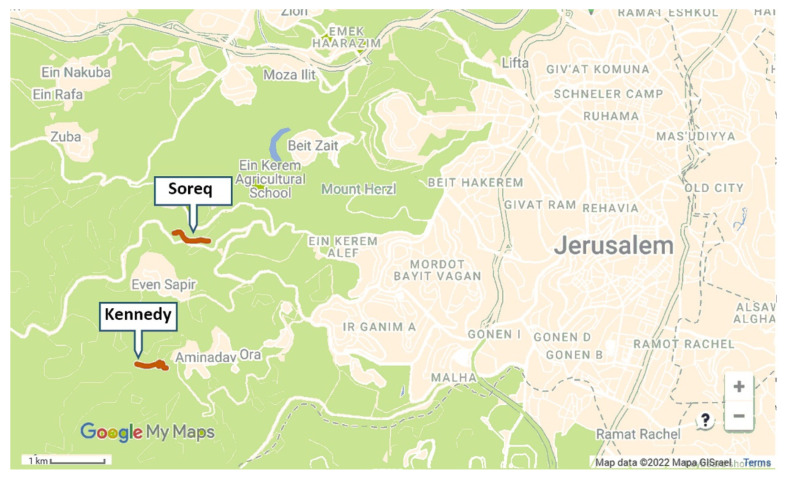
A map showing the locations of the Soreq and Kennedy study sites transects (red bold lines) relative to the western neighborhoods of Jerusalem. (Beige shading—built areas, green shading—open areas).

**Figure 2 insects-13-00816-f002:**
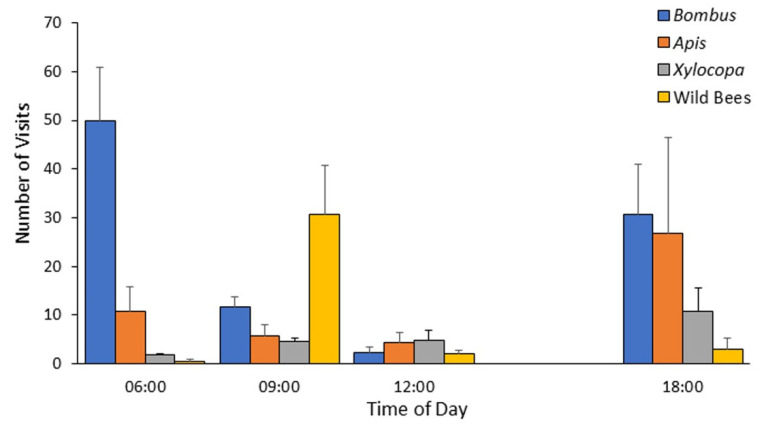
Mean number of visits per transect during four daily time points for bees classified into four categories during the peak season of 2012 (mid-May to mid-July) at the Soreq site. *Bombus*—*Bombus terrestris*; *Apis—Apis mellifera*; *Xylocopa*—various species of the genus *Xylocopa* (mostly *X. violacea*, and to lesser extent *X. pubescens* and *X. iris*); wild bees—all other bees not included in the three categories mentioned above. Error bars indicate standard errors.

**Figure 3 insects-13-00816-f003:**
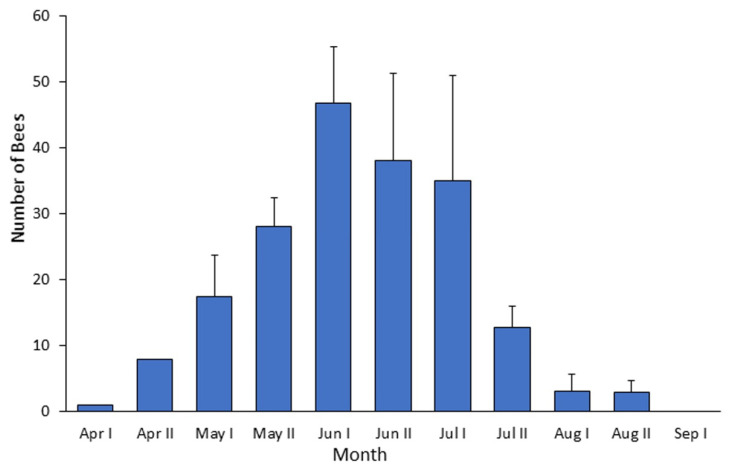
Mean number of *Bombus terrestris* records at the first transect after sunrise of each observation day, during the period from April to September at Soreq site. Values are the averages for the three sampling years; error bars depict standard errors between years.

**Figure 4 insects-13-00816-f004:**
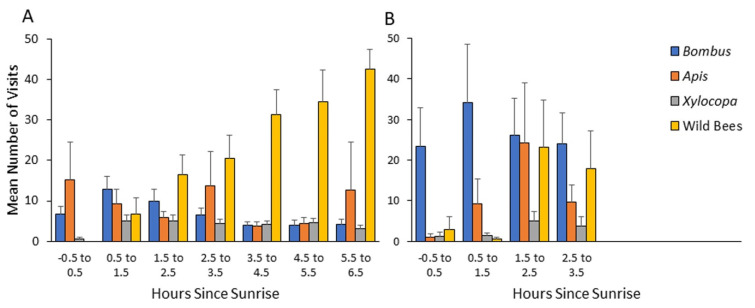
Mean number of visits per transect as a function of time after sunrise during 2013 (**A**) and 2014 (**B**) seasons at the Soreq site. *Bombus*—*Bombus terrestris*; *Apis—Apis mellifera*; *Xylocopa*—various species of the genus *Xylocopa* (mostly *X. violacea*, and to lesser extent *X. pubescens* and *X. iris*);.wild bees—all other bees not included in the three categories mentioned above. Bars indicate standard errors.

**Figure 5 insects-13-00816-f005:**
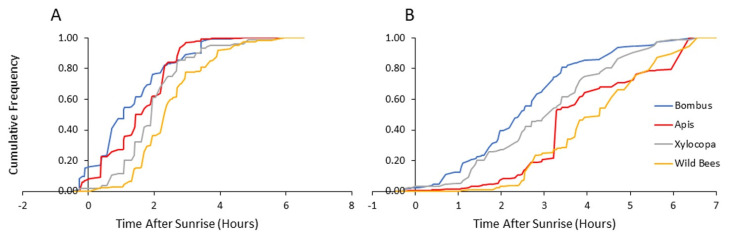
Flower visitation of bees from each category as a function of time after sunrise: (**A**) visitations to pollen-providing flowers and (**B**) visitations to nectar-providing flowers. The vertical axis presents the cumulative frequency of bee numbers for both 2013 and 2014.

**Figure 6 insects-13-00816-f006:**
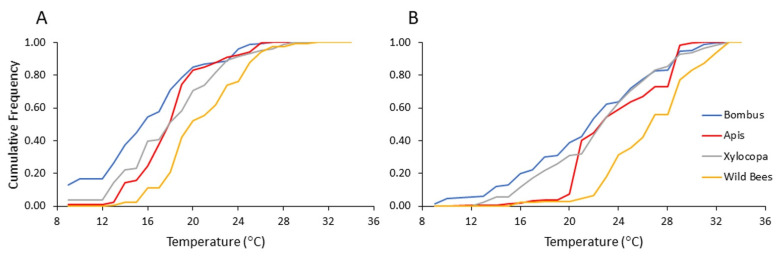
Flower visitation of bees from each category as a function of the ambient temperature (**A**) on pollen-providing flowers and (**B**) on nectar-providing flowers. The vertical axis presents the cumulative frequency of bee numbers recorded during 2013 and 2014.

**Figure 7 insects-13-00816-f007:**
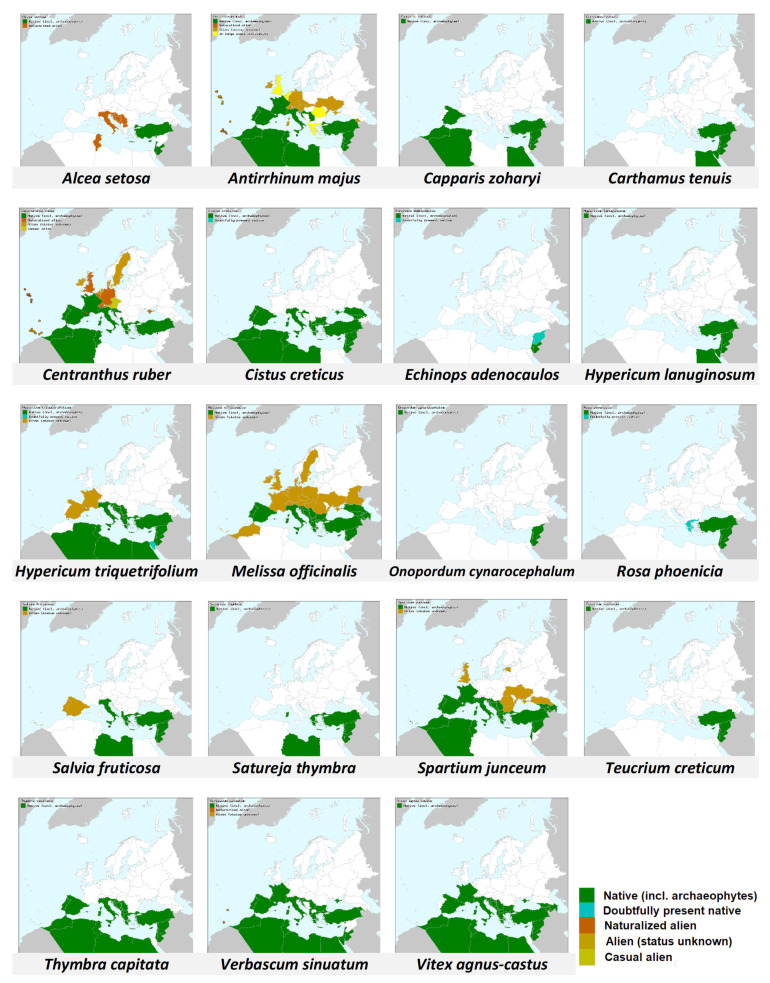
Distribution in Europe and the Mediterranean basin (color filling by countries) of the plant species visited by *B. terrestris* at the Soreq site in our study. Map source: Euro+Med (2006). Dark Green color indicates native range.

**Table 1 insects-13-00816-t001:** Plants visited by bumble bees in our study (flowering period).

Plant Species	Plant Family	Flowering Period	Nectar Source	Pollen Source
*Satureja thymbra*	Lamiaceae	mid April–mid June	Main	
*Antirrhinum majus*	Plantaginaceae	mid April–mid June	Main	
*Teucrium creticum*	Lamiaceae	May–mid July	Main	
*Echinops adenocaulos*	Asteraceae	June–mid August	Main	
*Vitex agnus-castus*	Lamiaceae	June–mid September	Main	Secondary
*Melissa officinalis*	Lamiaceae	mid May–June	Main	
*Thymbra capitata*	Lamiaceae	July	Main	
*Alcea setosa*	Malvaceae	May	Main	
*Salvia fruticosa*	Lamiaceae	mid April–mid June	Main	
*Onopordum cynarocephalum*	Asteraceae	May	Main	
*Centranthus ruber*	Caprifoliaceae	June	Main	
*Carthamus tenuis*	Asteraceae	mid July–August	Secondary	
*Capparis zoharyi*	Capparaceae	May–July	Secondary	Main
*Cistus creticus*	Cistaceae	mid April–mid July		Main
*Rosa phoenicia*	Rosaceae	May–mid June		Main
*Verbascum sinuatum*	Scrophulariaceae	mid May–mid September		Main
*Hypericum lanuginosum*	Hypericaceae	May–mid June		Secondary
*Hypericum triquetrifolium*	Hypericaceae	July–mid August		Secondary
*Spartium junceum* ^†^	Fabaceae	May		Secondary

^†^ only *Bombus* queens (but not workers) and carpenter bees succeeded in handling *Spartium junceum* flowers.

**Table 2 insects-13-00816-t002:** Visitation records on all flowers: Bumblebees compared to bees from each of the three other categories as a function of time elapsed since sunrise.

	*p*-Value of *Bombus* Versus:		
Year	*Apis*	*Xylocopa*	Wild Bees
2013	0.178	**<0.001 ***	**<0.001 ***
2014	0.105	0.028	0.036
Both	0.101	**<0.001 ***	**<0.001 ***

* Asterisks denote significance at the 5% level (after a Bonferroni correction for multiple comparisons).

**Table 3 insects-13-00816-t003:** Foraging on pollen vs. foraging on nectar flowers: The mean time after sunrise for each of the four bee categories while foraging for nectar or for pollen, with the significance of the differences between nectar and pollen.

	*Bombus*			*Apis*			*Xylocopa*			Wild Bees		
Year	Pollen	Nectar	*p*-Value	Pollen	Nectar	*p*-Value	Pollen	Nectar	*p*-Value	Pollen	Nectar	*p*-Value
2013	1.30 h	2.87 h	**<0.001 ***	1.16 h	4.06 h	**<0.001 ***	2.16 h	3.16 h	**0.011 ***	2.72 h	4.31	**0.002 ***
2014	1.34 h	2.10 h	**0.008 ***	2.08 h	2.44 h	0.322	1.75 h	2.55 h	0.144	2.16 h	2.87 h	0.264
Both	1.33 h	2.56 h	**<0.001 ***	1.57 h	3.98 h	**<0.001 ***	2.02 h	3.07 h	**0.006 ***	2.53 h	4.22 h	**0.002 ***

* Asterisks denote significance at the 5% level.

**Table 4 insects-13-00816-t004:** Foraging activity of bumblebees vs. each of the other three categories as a function of time after sunrise.

	On Pollen Flowers			On Nectar Flowers		
	*p*-Value of *Bombus* Versus:			*p*-Value of *Bombus* Versus:		
Year	*Apis*	*Xylocopa*	Wild Bees	*Apis*	*Xylocopa*	Wild Bees
2013	0.610	0.023	**0.002 ***	0.061	0.103	**<0.001 ***
2014	0.070	0.193	0.058	0.276	0.155	0.248
Both	0.417	**0.015 ***	**<0.001 ***	0.049	0.064	**<0.001 ***

* Asterisks denote significance at the 5% level (after a Bonferroni correction for multiple comparisons).

**Table 5 insects-13-00816-t005:** Presence of bumblebees vs. each of the other three categories during the first hour after sunrise.

	On Pollen Flowers			On Nectar Flowers		
	During First Hour after Sunrise			During First Hour after Sunrise		
	*p*-Value of Bombus Versus:			*p*-Value of Bombus Versus:		
Year	*Apis*	*Xylocopa*	Wild Bees	*Apis*	*Xylocopa*	Wild Bees
2013	0.504	**<0.001 ***	**<0.001 ***	0.435	0.113	0.127
2014	**0.004 ***	0.018	**0.008 ***	0.227	0.506	0.510
Both	0.196	**<0.001 ***	**<0.001 ***	0.346	0.126	0.142

* Asterisks denote significance at the 5% level (after a Bonferroni correction for multiple comparisons).

**Table 6 insects-13-00816-t006:** Visitation records on all flowers: Bumblebees compared to bees from each of the three other categories as a function of ambient temperature.

	*p*-Value of *Bombus* Versus:		
Year	*Apis*	*Xylocopa*	Wild Bees
2013	0.149	**<0.001 ***	**<0.001 ***
2014	0.078	0.203	**0.005 ***
Both	0.075	**<0.001 ***	**<0.001 ***

* Asterisks denote significance at the 5% level (after a Bonferroni correction for multiple comparisons).

**Table 7 insects-13-00816-t007:** Foraging on pollen vs. foraging on nectar flowers: The mean ambient temperature for each of the four bee categories while foraging for nectar or for pollen, with the significance of the differences between nectar and pollen.

	*Bombus*			*Apis*			*Xylocopa*			Wild Bees		
Year	Pollen	Nectar	*p*-Value	Pollen	Nectar	*p*-value	Pollen	Nectar	*p*-Value	Pollen	Nectar	*p*-Value
2013	16.35°	21.85°	**<0.001***	17.67°	24.04°	**0.012 ***	19.07°	22.89°	**0.004 ***	21.77°	26.94°	**0.003 ***
2014	16.19°	19.04°	**0.013***	19.36°	20.42°	0.368	17.20°	18.97°	0.273	19.88°	24.89°	0.114
Both	16.25°	20.67°	**<0.001 ***	18.42°	23.89°	**0.012 ***	18.44°	22.34°	**0.003 ***	21.13°	26.79°	**0.001 ***

* Asterisks denote significance at the 5% level.

**Table 8 insects-13-00816-t008:** Foraging activity of bumblebees vs. each of the other three categories as a function of ambient temperature.

	On Pollen Flowers			On Nectar Flowers		
	*p*-Value of *Bombus* Versus:			*p*-Value of *Bombus* Versus:		
Year	*Apis*	*Xylocopa*	Wild bees	*Apis*	*Xylocopa*	Wild bees
2013	0.215	0.026	**<0.001 ***	0.193	0.077	**<0.001 ***
2014	0.052	0.281	0.030	0.266	0.520	0.068
Both	0.104	0.021	**<0.001 ***	0.154	0.090	**<0.001 ***

* Asterisks denote significance at the 5% level (after a Bonferroni correction for multiple comparisons).

## Data Availability

All data are provided in the text, Appendix A, or upon request from the authors.

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
