# Peer review of "Earlier Morning Arrival to Pollen-Rewarding Flowers May Enable Feral Bumble Bees to Successfully Compete with Local Bee Species and Expand Their Distribution Range in a Mediterranean Habitat"

_insects, 2022, doi:10.3390/insects13090816_

Round 1

Reviewer 1 Report

Review.

This is an interesting paper looking at the potential for feral, range expanding, B. terrestris to outcompete members of the native bee community because they are both numerous and able to forage earlier in the day (and/or at cooler temperatures) than native, mostly solitary bees. Resource competition between native and introduced bee species (chiefly honey bees and this particular bumble bee) may contribute to overall declines in native pollinator numbers. Therefore, the paper is of general interest.

While I favor eventual publication, I have several serious criticisms which I would like to see addressed before its acceptable. These are:

1. Abundance data for the four classes of bees (bumble bees, honey bees, Xylocopa spp., other native bees) are given only for the initial year of survey (2012, Fig. 3). For the later two years we get figures 4 and 5 which plot cumulative frequencies of each class as a function of time of day or temperature. But these frequencies are the proportion of all bees of that class seen and there is no information on how many of each class of bees were seen. This MUST be rectified. First, it will strengthen the inference of potential for competition if we have a feel for the numbers of bumble bees relative to other types. Second when a future researcher wants to use this study as a baseline and see how things have changed, they will need the numbers. 

2. The authors need to state whether classification of a floral resource as primarily pollen or nectar providing is done based on observations of how bumble bees treat these flowers or all bees. Primarily we need to know whether all bees treat these flowers in the same way.

3. It would be useful if you could use a single analysis to assess the effects of temperature and time (which are obviously highly correlated). So this would be akin to multiple regression. At the very least, the authors need to say why this wasn’t or couldn’t be done.

4. Please indicate somewhere whether there are any feral Apis mellifera in your study area. In my experience, people in areas where Apis is introduced are surprised to hear that feral populations are rare or absent in parts of what is considered their native range.

5. Please clear up what is meant by pollen replenishment within flowers. I think you mean that pollen is presented in such a way that not all of it can be taken by the first visitor (for example Lupinus). But replenishment implies production of more pollen once some/all is removed, which happens for nectar.

6. Very importantly, throughout the manuscript, evidence of resource overlap and pre-emption is taken as evidence of competition per se, rather than potential for competition. The authors need to exercise more care on this point. Nowhere is there evidence that bumble bees depress populations of other bees. Importantly also, there is no discussion of whether bee populations are known to be resource limited. My feeling on this point is that the degree to which these populations are resource limited is equivocal, or at least explicit demonstrations are rare or absent. The authors may disagree with that but whatever their opinion, discussion of this should be required along with necessary use of “potential” before “competition” in most/all cases.

7. The discussion seems to assume that coevolved pollinators provide better plant reproductive service than do introduced ones. What is the evidence for this? In general, such coevolution is quite diffuse as most pollinators are not particularly specialized, something network studies have repeatedly shown. I don’t think there is much hard evidence for this generalization, so the authors should make clear that this is speculative.

Below you will find a long list of corrections by line number, mostly small grammatical ones. The topics above are all mentioned below as well. I apologize for the repetition but it might be useful to know the line numbers that prompted the criticisms.

15. these should be those

24. networks, or the pollination network

52. South America?

54. in, not at

68. spread

68. not clear what "high adaptability" means. Are we talking propensity for evolutionary adaptation or, as I think you mean, ability to use a wide range of resources and find nest sites in a wide range of habitats, all of which are listed. Maybe just drop high adaptability.

76. and use garden plants?

80. fix both et. and Al.

89. apostrophe needed for terrestris

92. should be “relative”

93. reduce and deplete are synonyms. I think you mean reduce or entirely remove or deplete entirely

94. Instead of less adapted you might want to use  “less capable of active foraging at low temperatures”. Body size and social warming clearly play a huge role here. Small solitary bees in cooler environments in, say, somewhat more northerly locations in Europe likely also forage when its later/warmer.

98 replaced rather than produced

120. A few …..

122. from, not since

126. scheduled the transects is not clear. I presume you mean walked the transects at these various times of day.

129. period, not transect

148. Is characterization of species of flower as a pollen or nectar source primarily based on observations of B. terrestris and do the other pollinators use these flowers the same way (i.e. if bumble bees use it for pollen only, do other bees do the same?

150. approaches

155. in, not at

166. During, not At

182. I think you should drop “that are adapted to cold temperatures”. Could just be a function of body size and sociality. At any rate, you don’t need adaptation here.

231. Why can’t you analyze the effects of time and temperature together in one model?

238. during, not on

244. a few

261. spp.

264. native, not wild. Most people would consider feral B. terrestris to be wild. Of course, native species would include the western honey bee, though its now managed. So word choices are difficult.

278. Why no bar graphs?

302-303. What is the evidence that other wild bees or carpenter bees forage for pollen before nectar? I don’t think you present any.

308. There are tropical bumble bees, so confine this statement to Bombus terrestris and temper it a bit. Even within its "native" range, there are some fairly warm areas.

356. Citations for the first sentence of discussion?

358. numbers, not scales

368. Discusses castes or at least temporal separation of gynes and workers. But no data were presented on this.

373. expansion, not distribution

379. As before, diminish and deplete are synonyms

380. “by the flowers” should be “within a flower”

384. a cold, temperate…

398. should be “twilight”?

402. Why weren’t they included among Xylocopa? Seems unreasonable to exclude them.

405. Please indicate somewhere whether there are any feral honey bees in your study area

406. was, not were

412-414. Awkward sentence. Maybe break it up.

415. what flowers replenish pollen? Some dole out their pollen such that not all can be removed during a visit (e.g. Lupinus), but I am unaware of the production of more pollen after the removal of some, which happens for nectar in at least some species.

418. Abundance data are only presented for the first year.

421. classify

423. delete the

427. al.

No discussion of the degree to which bee populations are known to be resource limited

435. does benefit should be benefits

437. are honey bees “local”? Maybe native, unmanaged bees

439. “supports” not “does comply”

442. There is only one study cited but sentence says “both studies”.

444-446. Are you suggesting that native bees in Israel are incapable of co-existing with B. terrestris? The more reasonable suggestion would be that their population sizes might be reduced in its presence.

452. spread, not dispersal

454. Such a plant ….

456. Drop thus and make it a separate sentence. The increasing …..

460-464. What is the evidence that native "coevolved" pollinators generally provide better service to native plants than do introduced pollinators?

474. Do introduced pollinators really favor introduced plants and rob natives as a generalization?

477. Nice to see some humility here, but above you asserted these things as generally true.

488. Honey bees outside their native range often have broad niche overlap with the community of native bees because they are supergeneralists. There is lots of documentation on this.

501. displacement of other bees from whatever plant honey bees focus on also commonly reported and should be discussed (e.g. Magrach et al., but there are more).

505-7 If its not significant, nor important to your central focus, probably should not discuss.

509 identification of, not identifying

510. It should be made clear whether early or late foraging bees would suffer most from competition with bumble bees. In fact, it could be argued both ways.

512-515. Its clear that bees that forage on flowers not visited by bumble bees  will suffer less competition, but this has nothing to do with time of day. This discussion of how to identify which species might be most prone to negative effects from bumble bees needs tightening up.

517. Competitor implies that B terrestris reduces populations of native bees. This has not been shown. The data show foraging niche overlap.

525. Your B. terrestris are not managed

Author Response

Manuscript ID: insects-1834176

Dear Editor,

We were very pleased with the overall positive response of the two reviewers, and we thank them for rigorously reviewing our manuscript and for their helpful comments which help us improve our contribution. Below please find our responses to each of the points raised by the two reviewers.

Reviewer #1

This is an interesting paper looking at the potential for feral, range expanding, B. terrestris to outcompete members of the native bee community because they are both numerous and able to forage earlier in the day (and/or at cooler temperatures) than native, mostly solitary bees…

*** We were very pleased with the overall positive feedback from this reviewer and thank her/him for the many helpful comments and suggestions, which we address in detail below.

  1. Abundance data for the four classes of bees (bumble bees, honey bees, Xylocopa spp., other native bees) are given only for the initial year of survey (2012, Fig. 3). For the later two years we get figures 4 and 5 which plot cumulative frequencies of each class as a function of time of day or temperature. But these frequencies are the proportion of all bees of that class seen and there is no information on how many of each class of bees were seen. This MUST be rectified. First, it will strengthen the inference of potential for competition if we have a feel for the numbers of bumble bees relative to other types. Second when a future researcher wants to use this study as a baseline and see how things have changed, they will need the numbers.

*** Along with the reviewer comment, we added two new plots for the years 2013 and 2014 that are presented in a new figure (Figure 4 in the revised manuscript). These plots show abundance data presented as mean number of visits per transect as a function of hours since sunrise. We further revised other figures and tables, and we hope that they are clearer now.

  1. The authors need to state whether classification of a floral resource as primarily pollen or nectar providing is done based on observations of how bumble bees treat these flowers or all bees. Primarily we need to know whether all bees treat these flowers in the same way.

*** Classification to pollen and nectar plants is based on our observations as part of this particular study. In most plant species, there was a single type of reward that we observed to be almost exclusively collected by all bee visitors. Exceptions of this were Capparis zoharyi, which primarily served as pollen provider, but we also saw bees collecting nectar, and Vitex agnus-castus which was primarily nectar provider, but bees were occasionally collecting pollen as well. We revised the corresponding texts in M&M to clarify this point.

  1. It would be useful if you could use a single analysis to assess the effects of temperature and time (which are obviously highly correlated). So this would be akin to multiple regression. At the very least, the authors need to say why this wasn’t or couldn’t be done.

*** We agree with this suggestion, which we indeed considered. Indeed, time and temperature are positively correlated, albeit not very high (r=0.87 in 2013 but only 0.61 in 2014), so it could be of interest to analyze the combined effect of both. Unfortunately, the distributions are not normal and do not fit any other distribution that can be handled by generalized two-way analyses.

  1. Please indicate somewhere whether there are any feral Apis mellifera in your study area. In my experience, people in areas where Apis is introduced are surprised to hear that feral populations are rare or absent in parts of what is considered their native range.

 *** We did not see feral colonies within the study area, but we know of sporadic evidence for feral colonies in the Judean Hills. We now refer to this point in the Material and Methods lines 126-129.

  1. Please clear up what is meant by pollen replenishment within flowers. I think you mean that pollen is presented in such a way that not all of it can be taken by the first visitor (for example Lupinus). But replenishment implies production of more pollen once some/all is removed, which happens for nectar.

*** While talking in the original manuscript about pollen which is sometimes replenished, we referred both to the dispensing of a portion of pollen in each visit, as the reviewer mentioned, and also to cases where anthers in the plant dehisce gradually, creating a refill-like effect which prevents the first visitors from collecting all of the pollen in a flower. We accept your comment that replenishment is not the right term for this, but anyhow, following another comment by you, we decided to omit the paragraph including this statement. We further clarified this point at several places throughout the manuscript.

  1. Very importantly, throughout the manuscript, evidence of resource overlap and pre-emption is taken as evidence of competition per se, rather than potential for competition. The authors need to exercise more care on this point. Nowhere is there evidence that bumble bees depress populations of other bees. Importantly also, there is no discussion of whether bee populations are known to be resource limited. My feeling on this point is that the degree to which these populations are resource limited is equivocal, or at least explicit demonstrations are rare or absent. The authors may disagree with that but whatever their opinion, discussion of this should be required along with necessary use of “potential” before “competition” in most/all cases.

 *** This is an excellent point with which we agree. Accordingly, we tone down the text on competition throughout the revised manuscript.

  1. The discussion seems to assume that coevolved pollinators provide better plant reproductive service than do introduced ones. What is the evidence for this? In general, such coevolution is quite diffuse as most pollinators are not particularly specialized, something network studies have repeatedly shown. I don’t think there is much hard evidence for this generalization, so the authors should make clear that this is speculative.

 *** We accept this point and omitted or toned down the discussion on co-evolution.

Below you will find a long list of corrections by line number, mostly small grammatical ones. The topics above are all mentioned below as well. I apologize for the repetition but it might be useful to know the line numbers that prompted the criticisms. 

*** Thanks for spotting these typos and mistakes. We have fixed all of them. Those comments for which there is a need for more details are addressed below.

  1. not clear what "high adaptability" means. Are we talking propensity for evolutionary adaptation or, as I think you mean, ability to use a wide range of resources and find nest sites in a wide range of habitats, all of which are listed. Maybe just drop high adaptability.

*** We changed “high adaptability in” to “the capacity to exploit”

  1. and use garden plants?

*** Correct, we added it to our text

  1. Instead of less adapted you might want to use “less capable of active foraging at low temperatures”. Body size and social warming clearly play a huge role here. Small solitary bees in cooler environments in, say, somewhat more northerly locations in Europe likely also forage when its later/warmer.

*** fixed accordingly

98 replaced rather than produced

*** We changed the text to “can be continuously replaced”

  1. A few …..; 122. from, not since; 126. scheduled the transects is not clear. I presume you mean walked the transects at these various times of day.

*** Fixed

  1. Is characterization of species of flower as a pollen or nectar source primarily based on observations of B. terrestris and do the other pollinators use these flowers the same way (i.e. if bumble bees use it for pollen only, do other bees do the same?

*** See our reply to Comment #2 above.

  1. approaches; 155. in, not at; 166. During, not At; 182. I think you should drop “that are adapted to cold temperatures”. Could just be a function of body size and sociality. At any rate, you don’t need adaptation here.

*** Fixed

  1. Why can’t you analyse the effects of time and temperature together in one model?

*** We indeed considered this possibility. Please see our explanation in our reply to comment #3 above.

  1. during, not on; 244. a few; 261. spp.

*** Fixed

  1. native, not wild. Most people would consider feral B. terrestris to be wild. Of course, native species would include the western honey bee, though its now managed. So word choices are difficult.

*** We changed “wild” to “native”

  1. Why no bar graphs?

*** Given that we calculated the number of visits as a function of temperature or time after sunrise, our data is continuous and we find it more informative to present it as line plots.

302-303. What is the evidence that other wild bees or carpenter bees forage for pollen before nectar? I don’t think you present any.

*** This information is presented in Table 3A, and similarly in Table 3B for temperature. This information can be also inferred from the data presented in Figs 5 and 6 in the revised version (4 and 5 in the original submission). The trend was not significant for 2014, but was statistically significant for 2013 and for the pooled data of 2013+2014.

  1. There are tropical bumble bees, so confine this statement to Bombus terrestris and temper it a bit. Even within its "native" range, there are some fairly warm areas.

*** Done

  1. Citations for the first sentence of discussion?; 358. numbers, not scales

*** Following a comment made be Reviewer #2 we decided to remove the opening sentence.

  1. Discusses castes or at least temporal separation of gynes and workers. But no data were presented on this.

*** As detailed in our M&M, we performed most of the systematic observations during the time in which mostly workers and no queens are active. Our observations of queens/gynes were more sporadic and cannot be presented in a way comparable to our worker data.

  1. expansion, not distribution; 379. As before, diminish and deplete are synonyms; 380. “by the flowers” should be “within a flower”; 384. a cold, temperate…; 398. should be “twilight”?

*** Fixed

  1. Why weren’t they included among Xylocopa? Seems unreasonable to exclude them.

*** Xylocopa olivierii is ecologically distinct from other carpenter bees, in both nesting habits and foraging times, which precedes even those of bumble bees and honey bees. Their activity before sunrise made it difficult to precisely record their flower visitation. As elaborated in the text, the few observations of bees of this species do not change the overall pattern.

  1. Please indicate somewhere whether there are any feral honey bees in your study area

*** We now added this information in L126-129 in the M&M of the revised manuscript.

  1. was, not were

*** Fixed

412-414. Awkward sentence. Maybe break it up.

*** We decided to entirely remove this sentence which is not very important for our discussion

  1. what flowers replenish pollen? Some dole out their pollen such that not all can be removed during a visit (e.g. Lupinus), but I am unaware of the production of more pollen after the removal of some, which happens for nectar in at least some species.

*** See our reply to major comment #5 above.

  1. Abundance data are only presented for the first year.

*** See our reply to major comment #1 above.

  1. classify; 423. delete the; 427. al.

*** Fixed

No discussion of the degree to which bee populations are known to be resource limited

*** Given the length of our Discussion, and the lack of solid data on this, we tone down our discussion along with the reviewer’s comment

  1. does benefit should be benefits

*** Fixed

  1. are honey bees “local”? Maybe native, unmanaged bees

*** We changed the sentence to “…earlier than the native, unmanaged bees.”

  1. “supports” not “does comply”; 442. There is only one study cited but sentence says “both studies”.

*** Fixed

444-446. Are you suggesting that native bees in Israel are incapable of co-existing with B. terrestris? The more reasonable suggestion would be that their population sizes might be reduced in its presence.

*** In order to avoid confusion, we edited the first part of this sentence, and removed the later part. We hope that the current text is cautious enough.

  1. spread, not dispersal; 454. Such a plant ….; 456. Drop thus and make it a separate sentence. The increasing …..

*** Fixed

460-464. What is the evidence that native "coevolved" pollinators generally provide better service to native plants than do introduced pollinators?

*** We edited the entire sentence such that we do not speak on coevolution.

  1. Do introduced pollinators really favor introduced plants and rob natives as a generalization?; 477. Nice to see some humility here, but above you asserted these things as generally true.

*** We removed this text, which was indeed quite speculative.

  1. Honey bees outside their native range often have broad niche overlap with the community of native bees because they are supergeneralists. There is lots of documentation on this.

*** We removed this entire section from the revised manuscript such that our Discussion is more focused and less speculative.

  1. displacement of other bees from whatever plant honey bees focus on also commonly reported and should be discussed (e.g. Magrach et al., but there are more).

*** We added a few citations. However, given the length of our Discussion, we decided not to not elaborate further on this issue.

505-7 If its not significant, nor important to your central focus, probably should not discuss.

*** We agree and removed this sentence.

509 identification of, not identifying

*** Fixed

  1. It should be made clear whether early or late foraging bees would suffer most from competition with bumble bees. In fact, it could be argued both ways.; 512-515. Its clear that bees that forage on flowers not visited by bumble bees will suffer less competition, but this has nothing to do with time of day. This discussion of how to identify which species might be most prone to negative effects from bumble bees needs tightening up.; 517. Competitor implies that B terrestris reduces populations of native bees. This has not been shown. The data show foraging niche overlap.; 525. Your B. terrestris are not managed

*** We tone down the last paragraph of the Discussion and limited it to calling for additional studies on the effect of bumble bees and honey bees on native bees and flowers, without elaborating on the possible results of these suggested studies.

Reviewer 2 Report

Dear authors,

Congratulations on the study. It is interesting and brings background for conservation measures and future studies.

I have some suggestions and questions, especially about methods and some results. Please find the comments in the file attached.

Best regards.

Author Response

Dear Editor,

We were very pleased with the overall positive response of the two reviewers, and we thank them for rigorously reviewing our manuscript and for their helpful comments which help us improve our contribution. Below please find our responses to each of the points raised by the two reviewers.

Reviewer #2

L15. Do you mean native? If so, I suggest that you say "native bee species"; Do you mean all bee species?

*** Fixed

L103. The season mentioned was identified during the first year of observation and after that the observations occurred only when bumble bees were active, is this it? To leave no doubts, I suggest that you explain about the first year and then the other 2 years.

*** We edited the sentence and hope that it is clear now

L133. Male is not a caste. Correct this, please.

*** Fixed

L139. I think it doesn't need to be in capital letters, because it is not the name of the order, e.g. Diptera.

*** Fixed

L171. This table can be organized. I suggest that you include the following columns: plant species, plant family, flowering period, nectar source (main or secondary), pollen source (main or secondary).

*** Done.

L176. This figure can be improved. It is hard to see the light gray. What is the light gray? Is the green vegetation? Where are the limits between Soreq and Kennedy? Is there a better shape than those ballons to indicate the sites, e.g. arrows? Note that figure legends should be independent of the text, so I suggest that the coordinates of sites and transects are informed here.

*** Done. Including using a different map format.

L240. Do you mean that the bee species used only pollen or nectar of a certain plant species or that an individual bee did not collected both of one flower?

*** We slightly edited the text and hope that our meaning is now clear.

L243. Is this an extra observation that was not described in the material and methods? I don't understand why this sentence is here in the results.

*** Following the reviewer’s comment, we now moved this sentence to the M&M

L311. When I read the material and methods it was not clear to me if the temperature was measured only at the beginning of the first transect or at the beginning of each transect. Please clarify.

*** We edited the corresponding M&M section (new L136-137) which now reads: “At the beginning of each sampling walk we recorded the ambient temperature using Casella mercury-filled thermometer.”

L366. I would highlight the results from this study first instead of repeating why this study was done.

*** Along with this comment, we decided to remove the first sentence of the Discussion which repeats information already presented in the Introduction

Fig. 6 legend. What is the beige?

*** We now improve the figure and added a legend with the different color scales (now figure 7).

Round 2

Reviewer 1 Report

Authors seem to have done a good job responding to previous criticism and appropriately modifying the text. My one current criticism has to do with the current title. Title is no longer grammatical. I am fine with "outcompete" because "may" acts as an appropriate qualifier. If you use "successfully compete" you need "with" or "against" local bee species and "expand their geographical range in Mediterranean habitat" would be better than "distribution range".